# Competing Endogenous RNA Regulatory Networks of hsa_circ_0126672 in Pathophysiology of Coronary Heart Disease

**DOI:** 10.3390/genes14030550

**Published:** 2023-02-22

**Authors:** Muhammad Rafiq, Abdullahi Dandare, Arham Javed, Afrose Liaquat, Afraz Ahmad Raja, Hassaan Mehboob Awan, Muhammad Jawad Khan, Aisha Naeem

**Affiliations:** 1Department of Biosciences, COMSATS University Islamabad, Islamabad 45550, Pakistan; 2Department of Biochemistry, Shifa College of Medicine, Shifa Tameer-e-Millat University, Islamabad 45550, Pakistan; 3Department of Biochemistry, Usmanu Danfodiyo University Sokoto, Sokoto P.M.B 2346, Nigeria; 4Health Research Governance Department, Ministry of Public Health, Doha P.O. Box 42, Qatar; 5Department of Oncology, Lombardi Comprehensive Cancer Center, Georgetown University Medical Center, Washington, DC 20057, USA

**Keywords:** circRNA, miRNAs, gene expression, biomarker, CHD, atherosclerosis, ceRNA

## Abstract

Coronary heart disease (CHD) is a global health concern, and its molecular origin is not fully elucidated. Dysregulation of ncRNAs has been linked to many metabolic and infectious diseases. This study aimed to explore the role of circRNAs in the pathogenesis of CHD and predicted a candidate circRNA that could be targeted for therapeutic approaches to the disease. circRNAs associated with CHD were identified and CHD gene expression profiles were obtained, and analyzed with GEO2R. In addition, differentially expressed miRNA target genes (miR-DEGs) were identified and subjected to functional enrichment analysis. Networks of circRNA/miRNA/mRNA and the miRNA/affected pathways were constructed. Furthermore, a miRNA/mRNA homology study was performed. We identified that hsa_circ_0126672 was strongly associated with the CHD pathology by competing for endogenous RNA (ceRNA) mechanisms. hsa_circ_0126672 characteristically sponges miR-145-5p, miR-186-5p, miR-548c-3p, miR-7-5p, miR-495-3p, miR-203a-3p, and miR-21. Up-regulation of has_circ_0126672 affected various CHD-related cellular functions, such as atherosclerosis, JAK/STAT, and Apelin signaling pathways. Our results also revealed a perfect and stable interaction for the hybrid of miR-145-5p with *NOS1* and *RPS6KB1*. Finally, miR-145-5p had the highest degree of interaction with the validated small molecules. Henchashsa_circ_0126672 and target miRNAs, notably miR-145-5p, could be good candidates for the diagnosis and therapeutic approaches to CHD.

## 1. Introduction

A total of 17 million annual fatalities worldwide, representing about 30% of all deaths, are caused by cardiovascular diseases (CVDs). The most typical type of CVD is coronary heart disease (CHD) or ischemic heart disease (IHD), which accounts for about 38% of cardiovascular deaths in women and 46% in men [1]. The complex biology of CHD commences with endothelial dysfunction and chronic inflammation in the coronary arteries. The formation and subsequent development of atherosclerotic plaques within the coronary arteries restrict the blood supply to the heart and induce myocardial ischemia [2]. It was generally accepted that CHD is a major global problem that is more widespread in South Asian nations, e.g., Sri Lanka, Bangladesh, India, Nepal, Afghanistan, and Pakistan, compared to other countries [3,4]. This prevalence could be associated with genetics and unhealthy lifestyles [5]. Approximately 40–60% of the risks for coronary artery disease are caused by genetic predisposition [6]. Sedentary lifestyle, smoking, excessive alcohol consumption, unhealthy diet, oxidative stress, obesity, elevated serum cholesterol, hypertension, and diabetes are also established risk factors for CHD and powerful predictors of the variation in disease rates among populations [7]. It is therefore deduced that CHD is affected by hereditary and environmental factors [8].

Gene expression profiles have reflected pathological states in several disorders, e.g., cancer [9], chronic kidney disease (CKD) [10], metabolic syndrome [11], and CHD [12]. Scientific investigation indicates the gradual changes in gene expression profile during the development of CHD [12]. The use of gene expression signatures to discover CHD biomarkers for diagnosis, treatment, prognosis, and monitoring of the disease has yielded promising results [13,14]. Pro-platelet basic protein (PBP) and α-defensin (DEFA1/DEFA3) were identified as potential biomarkers of CHD in the Thai population [15]. The considerable effects of the pro-protein convertase subtilisin/kexin type 9 (*PCSK9*) gene on the low-density lipoprotein receptor (LDLR) and ultimately the plasma level of LDL-cholesterol are suggested as a good candidate for the therapy of CHD [16]. In addition, C-reactive protein (CRP) was identified as a powerful inflammatory biomarker in CHD [17].

Additionally, non-coding RNAs, e.g., long-chain non-coding RNAs (lncRNA), micro RNAs (miRNAs), and circular RNAs (circRNAs), play a crucial role in the regulation of more than half of protein-coding transcripts and are implicated in the regulation of almost every biological process within the cellular environment [18]. Thus, they play a pivotal role in the pathology of several diseases, e.g., neurodegenerative diseases [19], metabolic syndrome [20], and cancer [21]. Despite the extensive information regarding CHD, the disease is still prevalent in low- and middle-income countries. Hence, research on the role of ncRNAs as a biomarker of the disease is necessary as it is not fully elucidated. This study helped in the prediction of novel molecular marker(s) and could be utilized as either a therapeutic target and/or diagnostic biomarker of the disease.

## 2. Materials and Methods

### 2.1. Search for circRNAs Associated with CHD and Identification of Their Target miRNAs

circRNAs were identified by a comprehensive literature search and the circRNA2Disease database [22]. In five separate experiments, a total of 116 circRNAs associated with CVD were obtained [23,24,25,26,27] (Appendix A). These circRNAs were individually submitted to a computational tool known as the “Circular RNA Interactome” that helps in the mapping and prediction of binding positions for particular miRNAs on reported circRNAs [28]. Three circRNAs; has_circ_0092576, has_circ_0078837, and has_circ_0126672, were selected for further study based on the abundance of miRNA binding sites the circRNAs possessed (Appendix A). It was speculated that the higher the number of binding sites, the more circRNA could efficiently regulate the target miRNAs. The circRNAs possessing less than 5 binding sites for any miRNA were not captured in either the Appendix A or the presented data.

### 2.2. Search for CHD Differentially Expressed Genes (DEGs)

Gene expression raw data profiles by array were obtained from the GEO datasets of the NCBI. Only peripheral blood samples with CHD and healthy control data were selected for further analysis (Table 1). These experiments were submitted to GEO2R for statistical analysis by using already reported methodology [29] to acquire CHD differentially expressed genes (DEGs) and their expression pattern. The cutoff value for statistical significance was set at *p* < 0.05. The common differentially expressed genes (cDEGs) across the experiments that satisfied the predefined criteria were sorted by the online tool Bioinformatics and Evolutionary Genomics (http://bioinformatics.psb.ugent.be/webtools/Venn/, accessed on 20 April 2022). The fold changes of each gene obtained from the selected experiments were averaged and assigned to the corresponding gene (Appendix A).

### 2.3. Identification of Differentially Expressed miRNA Target Genes (miRNA-DEGs)

The miRNA database was queried for a list of predicted target genes for each miRNA [30]. Using the Bioinformatics and Evolutionary Genomics tool, the predicted miRNA target genes were compared to cDEG. The genes present across these two lists were designated as differentially expressed miRNA target genes (miR-DEGs) [20,31] (Appendix A).

### 2.4. Functional Enrichment Analysis

The database for annotation, visualization, and integrated discovery (DAVID) version 6.8 was employed for the functional enrichment study [32]. This enables a thorough knowledge of the biological significance of a given list of genes and establishes a relationship between the genes and associated illnesses or disorders. The common differentially expressed genes between all five experiments were sorted. Overlapping cDEGs and miRNA target genes (miR-DEGs) were separately identified for each miRNA and individually submitted to DAVID for analysis, setting *Homo sapiens* as the reference species. The tool first associates each gene identification with a gene ontology (GO) phrase classified into three broad categories: biological processes, molecular activities, and cellular compartments. On the other hand, the Kyoto encyclopedia of genes and genomes (KEGG) was utilized for pathway annotations of miR-DEGs [33].

### 2.5. Network Illustrations

The network interactions of circRNAs/miRNAs, and circRNA/miRNA/pathways, as well as circRNA/miRNA/gene involved in the regulation of biological processes relevant to CHD were generated using Cytoscape (version 3.8.1) [34].

### 2.6. miRNA and Target Sequence Homology Study

FASTA sequences for each miR-DEGs implicated in the control of the JAK/STAT and Apelin signaling pathways were obtained from the NCBI website. The sequences of mature miRNAs were retrieved from the miRDB (http://mirdb.org/ontology.html/, accessed on 20 May 2022). The nucleotide sequence of miRNA and predicted target DEGs were subjected to the BiBiserv RNA hybridization tool for the homology study [35]. The minimum free energy (MFE) value −15 kcal/mol was used as a cutoff value (threshold) [36]. The MFE value of −27 kcal/mol is comparable to the MFE for one of the four best hybridizations of let-7 [35]. Thus, an MFE of ≤−27 kcal/mol was used to predict good hybridization between miRNA and target mRNA. Although, MFE of ≤−30 kcal/mol is considered as perfect hybridization [37].

### 2.7. miRNA and Small Molecules Interaction

In order to investigate the potential of miRNAs as pharmacogenomics biomarkers, small molecule-miRNA network-based inference (SMiR-NBI), assessable at (http://lmmd.ecust.edu.cn/database/smir-nbi/, accessed on 12 June 2022) was employed to identify the experimentally validated small molecule that could interact with the miRNA to inhibit or enhance its expression [38]. The SM-miR network was constructed using the Cytoscape software. The degree of connectivity between the miRNAs and small molecule/drugs was analyzed with the same software [34]. The method employed for the selection and analysis of data was summarized as a flowchart (Figure 1).

## 3. Results

### 3.1. Identification of Differentially Expressed Genes and Selected Circular RNAs

We have recently published the role of the hsa_circ_0092576 regulatory network in the pathogenesis of CHD [39]. Herein, we emphasize the detailed information about the competing endogenous RNA regulatory network of hsa_circ_0126672 in the pathophysiology of CHD. A total of 13 gene expression array experiments of CHD were identified from NCBI websites (Figure 1). However, only five experiments satisfied the inclusion criteria, thus included in the analysis (Table 1).

In order to have a quick visualization of the pattern and degree of expression of genes as well as significantly dysregulated genes, volcanic plots for the GEO datasets of the five included experiments (GSE71226, GSE12288, GSE56885, GSE42148, and GSE20681) were constructed (Figure 2).

In addition, a total of five experiments on circRNAs were identified, in which 116 circRNAs were differentially expressed (Appendix A). However, only three circRNAs were selected for further analysis, considering their potentials in regulating the target miRNAs due to the high binding sites to the target miRNA (Appendix A).

The number of dysregulated genes in all five experiments was illustrated by a Venn diagram. Genes present in at least 3 of these experiments were considered common DEGs (cDEGs) (Figure 3A). A total of 566 cDEGs were found and thus considered for further computational analysis. The calculated average fold changes of the cDEGs (Appendix A) show that a total of 38% of the genes were down-regulated, compared to 62% that were up-regulated (Figure 3B).

### 3.2. Interaction of Circular RNA with the Selected miRNAs

A network of circRNAs and target miRNAs with more than 10 binding sites on at least one of the presented circRNAs was illustrated (Figure 4). It showed that 60 miRNAs were targeted by at least two circRNAs. Conversely, 29, 17, and 7 miRNAs were uniquely regulated by hsa_circ_0126672, hsa_circ_0078837, and hsa_circ_0092576, respectively (Figure 4A). It was observed that a total of 85 miRNAs interact with hsa_circ_0126672, of which 7 of the miRNAs (miR-548c-3p, miR-495-3-p, miR-186, miR-203, miR-21-5p, miR-7-5p, and miR-145-5p) were selected for further analysis (Figure 4B).

Several differentially expressed miRNA target genes were presented (Figure 4). It was observed that miR-548c-3p had the highest number of target DEGs, whereas miR-203a-3p had the least number of target genes (Appendix A). The number of up-regulated target genes was higher than the number of down-regulated target genes for all these miRNAs by at least two folds.

### 3.3. Gene Ontology and KEEG Pathway Analysis

In order to study the biological function impacted by the dysregulation of selected circRNAs and their corresponding target miRNAs, functional enrichment analysis for miR-DEGs of each of the miR-7-5p, miR-548c-3p, miR-21-5p, miR-186-5p, miR-145-5p, miR-203a-3p, and miR-495-3p was conducted (Appendix A).

The functional enrichment analysis enabled the discovery of enriched biological themes, notably gene ontology (GO) terms categorized into three primary sections: biological processes, cellular compartments, and molecular functions (Figure 5).

All the miRNAs examined in this work were shown to have a substantial role in many essential biological processes linked with CHD diseases, including transcription regulation from the RNA polymerase II promoter and DNA template, cell proliferation, and gene expression (Figure 5A). miR-548c-3p was shown to be solely engaged in circadian rhythm and PI3MAPK regulation.

Additionally, the findings demonstrated the essential role of miRNAs in the regulation of genes involved in molecular functions. miR-DEGs were shown to have a significant role in protein binding, poly (A) RNA binding, ATP binding, metal ion binding, and nucleic acid binding (Figure 5B). Each of these miRNAs had a role in the regulation of genes involved in metal ion binding. The transcription factor binding and fibroblast growth factor binding miRNAs controlled the fewest genes. Additionally, miR-548c-3p and miR-203a-3p were shown to be involved in the regulation of genes related to fibroblast growth factor. The third GO term is a cellular compartment that defines the location in which a gene product executes its biological tasks. The findings of this research demonstrated that all tested miRNAs had a substantial impact on the regulation of genes whose products perform biological tasks inside the nucleoplasm, nucleus, and cytoplasm compartments (Figure 5C).

### 3.4. Circular RNA Networks

In order to have a clear view of the participation of selected circRNAs involved in CHD-related biological processes as predicted in this research, a regulatory network of circRNA/miRNA/mRNA was constructed (Figure 6A). A total of 124 miR-DEGs were present in the CHD-related biological processes, as shown in Figure 6A. The number of up-regulated miR-DEGs was higher than the number of down-regulated miR-DEGs by three folds, with an approximate percentage composition of 76% and 24%, respectively. Similarly, a total of 74 dysregulated genes in biological processes were targets of at least two miRNAs; however, 50 genes were uniquely regulated by different miRNAs.

The regulatory role of the has_circ_0126672 targeting miRNAs participating in the CHD-related pathways was further investigated by using KEGG pathway analysis (Appendix A). It was revealed that many genes associated with CHD related signaling pathways were regulated via miRNAs (Figure 6B). TGF-β, relaxin, apelin, Hippo, JAK/STAT, MAPK, VEGF, mTOR, and PPAR are examples of these pathways. Insulin resistance and atherosclerosis were among the pathological diseases anticipated to be related to the dysregulation of hsa_circ_0126672 or its target miRNAs. Additionally, cardiac muscle contraction, the complement coagulation cascade, aldosterone synthesis and secretion, platelet activation, cellular senescence, renin secretion, and circadian rhythm were all impacted. The JAK/STAT and Apelin signaling pathways are crucial in CHD pathology and are regulated by a large number of miR-DEGs; thus, it is presented to visualize the genes involved as well as their pattern of expression. All miR-DEGs except *JAK2* were up-regulated in JAK/STAT (Appendix A) and *APLNR* in Apelin signaling pathways (Appendix A).

### 3.5. Homology Study and miRNAs Interaction with Small Molecules

The results of miRNA and target gene homology studies revealed that all duplexes had an MFE value less than the threshold (−15 kcal/mol) (Appendix A). However, only the hybrids with an MFE value of ≤−27 kcal/mol were presented (Appendix A). The hybrid of miR-145-5p and three of its target genes; *RPS6KB1*, *NOS1,* and *APLNR,* had perfect and stable interactions with MFE values of −38.8, −31.4, and −34.9 kcal/mol, respectively. The binding patterns of the aforementioned duplexes are good enough to predict stable interactions between miR-145-5p and its target genes (*RPS6KB1*, *NOS1*, and *APLNR)*.

### 3.6. Interaction between miRNAs and Small Molecules

A comprehensive network connecting miRNAs and small molecules was constructed (Figure 7). The expression of miRNAs was affected by many drugs. The degree, otherwise called connectivity counts, was analyzed.

miR-7-5p had the highest degree with 12 edges, followed by miR-203a-3p with 11 edges, and then miR-145-5p with 9 edges (Appendix A), hence they were the most studied miRNAs. miR-7-5p had an equal number of positive and negative regulators. It was noticed that miR-145-5p can be positively regulated by eight different small molecules and negatively regulated by only one small molecule. Vorinostat is presently the only drug deposited in the SMiR-NCBI database that has been validated to regulate the expression of miR-548c-3p. The expression of certain miRNAs was significantly affected by the combination of a few drugs. For instance, the combination of 5-aza-2′-deoxycytidine and trichostatin-A or 4-phenyl butyric acid and 5-aza-2′-deoxycytidine showed an up-regulated effect on miR-495-3p.

## 4. Discussion

Non-coding RNAs (ncRNAs) play a vital role in the regulation of physiological processes by regulating gene expression at both transcriptional and translational levels [2]. Aberrant ncRNAs’ expression is one of the underlying mechanisms that leads to the initiation and progression of several diseases, including cancer, CVD, obesity, and diabetes mellitus [40,41]. The present study predicted that ncRNAs are crucial for the regulation of genes involved in cellular processes linked to CHD. hsa_circ_0126672 had played a role in the onset and development of CHD by sponging its target miRNAs, e.g., miR-7-5p, miR-548c-3p, miR-21-5p, miR-186-5p, miR-145-5p, miR-203a-3p, and miR-495-3p. Numerous studies have shown a relationship between the expression of circRNAs and CVDs [27,42,43,44,45]. The number of up-regulated genes in CHD could be the outcome of the overexpression of has_circ_0126672. The miRNA sponge effect eradicates its impact on gene expression, resulting in the overexpression of the target genes [20,46]. Instead of miRNA binding to the target genes and inhibiting their expression, it can preferentially bind to the circRNA, thus favoring the expression of the miRNA target genes. Additionally, the up-regulation of has_circ_0126672 may account for the up-regulation of many cellular functions, including JAK/STAT and Apelin pathways, observed in this study.

Based on the functional enrichment study of miR-DEGs, the pathogenesis of CHD is governed by miR-7-5p, miR-548c-3p, miR-21-5p, miR-186-5p, miR-145-5p, and miR-203a-3p, and miR-495-3p. A significant number of differentially expressed genes targeted by these miRNAs participated in many biological processes that are either directly or indirectly linked to CHD, including protein phosphorylation, regulation of gene expression, circadian rhythm, production of vascular endothelial growth factor, and cell proliferation. In addition, many relevant signaling pathways e.g., Apelin, JAK/STAT, MAPK, PI3K, AKT, TGF-β, mTOR, VEGF, FoXO, Relaxin, and PPAR signaling pathways were affected. Other dysregulated signaling pathways associated with CHD included cardiac muscle contraction, insulin resistance, atherosclerosis, vasopressin-regulated water reabsorption, aldosterone synthesis and release, and platelet activation.

Protein phosphorylation is an important cellular process necessary for normal homeostasis and development. Dysregulation of protein phosphorylation leads to many types of diseases, including cardiovascular diseases [47,48]. Human heart failure, which could result from CHD, has been related to the limited extent of phosphorylation of thin-filament proteins [48]. The regulation of protein phosphorylation was under the control of miR-145-5p, miR-21-5p, miR-495-3p, and miR-7-5p via their interactions with target genes involved in the process. Except for miR-21-5p, the present study predicted the participation of all these miRNAs in the regulation of cell proliferation. This is in accordance with the previous report of Liu et al., [49] who highlighted the regulatory effect of miR-23 in cell proliferation and apoptosis of vascular smooth muscle cells in CHD. Recently, the role of miR-7-5p as a biomarker of CHD was ascertained. However, down-regulation was associated with a high risk of developing CHD, thus miR-7-5p may be involved in the pathophysiology of the disease [50]. Our study showed significant involvement of miR-7-5p in CHD pathology via regulating numerous signaling pathways, including cell proliferation, protein phosphorylation, and vascular and endothelial growth factors production.

Contrary to our findings, miR-21 enhanced the cell proliferation of murine cardiac stem cells post-myocardial infarction via the inhibition of *PTEN* expression and stimulation of the PI3K/Akt pathway [51]. One of the underlying mechanisms of human health is circadian rhythm [52]. It is critically important in cardiovascular physiology. Disturbance in circadian rhythms has been associated with a high risk of acquiring cardiac complications and harmful cardiovascular incidents [53]. Our findings showed miR-548c-3p targets DEGs that are involved in circadian rhythm; thus, it was proposed that miR-548c-3p has a vital role in the physiology of CVD via regulation of *NRIP1*, *NAMPT*, *CREB1*, *NAMPT,* and *ID3*. Vascular endothelial growth factor (VEGF) plays an integral part in angiogenesis, vascular pathology, and atherosclerosis and has been associated with CVDs. Thus, VEGF possesses positive and negative effects in CHD [54]. The present study also predicted the involvement of miR-7-5p and miR-548c-3p in the generation of VEGF by targeting the *IL6ST*, *HIF1A,* and *PTGS2*. This may be considered a different fundamental mechanism through which hsa_circ_0126672 and their target miRNAs, miR-7-5p, and miR-548c-3p, have been associated with the pathogenesis of CHD. Previously, miR-7-5p was determined to be an effective molecule for the regulation of VEGF. The inhibition of miR-7-5p resulted in increased angiogenesis via the up-regulation of VEGF by the direct target of Krüppel-like factor 4 (KLF4) [55]. Similar to our findings, the role of miR-203a-3p in the regulation of VEGF secretion was reported [56,57]. Therefore, it was predicted that the down-regulation of miR-7-5p caused by the up-regulation of has_circ_0126672 would promote the production of VEGF in CHD patients. Among predicted miRNAs, the miR-548c-3p exhibited a greater number of target genes and showed strong relation with cellular functions implicated in the pathophysiology of CHD. The function of miR-548c-3p in CHD was not well elaborated in previous studies. However, its strong association with pulmonary hypertension was reported [58].

The KEGG pathway analysis demonstrated the function of non-coding RNA (ncRNA) in the regulation of several signaling pathways and important physiological processes associated with CHD. The present work emphasized the JAK/STAT and Apelin signaling pathways due to their relevance in the pathology of CVDs and a higher degree of interaction with the analyzed miRNAs, as well as a relatively large proportion of miR-DEGs dysregulated in these pathways. It is generally accepted that the JAK/STAT pathway is involved in several cardiac pathologies [59]. It was believed that acute stimulation of the JAK/STAT signaling pathway is protective for cardiac cells, while persistent stimulation of the pathway can result in heart failure [60]. The contribution of a large number of miR-DEGs was depicted in this study. This reflects the regulatory function of the JAK/STAT signaling pathway by the hsa_circ_0126672 and its target miRNAs, notably miR-7-5p, miR-548c-3p, miR-21-5p, miR-186-5p, miR-145-5p, and miR-203a-3p and miR-495-3p. Previous reports have demonstrated the involvement of ncRNAs in the regulation of the JAK/STAT signaling pathway [61,62]. Our findings are supported by Li and Zeng [63], who described the effective regulatory effects of miR-21 on the JAK-STAT signaling pathway via the suppression of *STAT3* in juvenile idiopathic arthritis patients. Conversely, in another report, it was suggested that miR-21 and miR-9a may take part in JAK/STAT signaling pathway activation [64]. Our results are also in agreement with the previous report, which stated that the miR-548c-3p regulated genes are potent modulators of pathways associated with tumor development and metastasis, including the JAK-STAT signaling pathway [65]. In addition, it was cited that miR-145 could significantly regulate the expression of genes associated with the JAK/STAT pathway [66]. This study revealed that overexpressed genes were significantly involved in atherosclerosis and the JAK/STAT pathway. Thus, the activation of these pathways was predicted to support the fact that the onset and progression of atherosclerosis and hypertension largely depend on the activation of the JAK/STAT signaling system [67,68].

Our data highlighted that a large number of miR-DEGs were implicated in the control of the Apelin signaling pathway, which is one of the crucial signaling transduction pathways and is considered an important pathway involved in cardiovascular homeostasis [69]. The Apelin signaling pathway provokes several physiological mechanisms and processes, e.g., cardiac contractility, blood pressure regulation, angiogenesis, the endocrine stress response, energy metabolism, and fluid homeostasis. This pathway also contributes to the pathogenesis of several diseases, including obesity, diabetes, heart diseases, and many forms of cancer [69,70,71]. It was previously reported that Apelin signaling may serve as a key atherosclerosis-protective marker to control the onset of CHD [72]. The miR-DEGs involved in the regulation of the Apelin signaling pathway highlighted the implication of hsa_circ_0126672 in CHD pathogenesis. The up-regulated circRNA sponges their target miRNAs, thereby competitively hindering the miRNA’s suppression influence on its target gene, which resulted in the up-regulation of the genes, as observed in the present research. The effect of miRNAs in the regulation of Apelin or the Apelin signaling pathway was previously reported. Zhou et al. [73] reported that miR-195 inhibited the development of lung cancer via the regulation of Apelin expression. In another report, miR-503 enhanced angiotensin II-induced cardiac fibrosis via the target of the *Apelin 13* gene [74]. Similar to our findings, an increase in the expression of Apelin in the atherosclerotic coronary artery was reported [75]. The higher level of Apelin may have a valuable or adverse impact on the progression of atherosclerosis. The negative consequences may include atherosclerosis and oxidative stress [76]. In contrast, Apelin has a positive effect by decreasing angiotensin II, thereby reducing atherosclerosis [77].

Homology studies between miRNA and mRNA enabled the determination of the pattern of interaction between miRNA and its target gene, as well as the MFE needed for the duplex to form. This is critical for identifying the potential of miRNAs to effectively bind to and control the target expression of the target sequence. Predicting and validating miRNA-target interactions is critical for comprehending miRNA’s involvement in complex networks that regulate cellular activities [78]. In the present study, eight miRNA:mRNA hybrids with good binding affinity and perfect or nearly perfect interaction were presented. These characteristics serve as a predictor of the regulatory potential of miRNA for its target genes. The alignment in Watson and Crick matching is considered perfect if there is no gap observed between the miRNA and target gene sequence [79]. The duplexe miR-186-5p:*GNB4*, miR-186-5p:*IL6ST*, miR-145-5p:*GNAQ*, miR-145-5p:*SOCS2*, miR-145-5p:*RPS6KB1*, miR-145-5p:*GABARPL1*, miR-145-5p:*APLNR*, and miR-145-5p:*NOS1* are relatively good, with MFE results of −27 kcal/mol, which are identical to the MFE results for the hybridization of the let-7 and the 3’UTR of *Caenorhabditis elegans*. One of the most well-known duplexes of CELF35-1 and let-7 [35]. According to a study, if the MFE value is below −30 kcal/mol, the miRNA:mRNA duplex will execute a stable and efficient interaction [37]. In addition, the lower the MFE value, the better the binding affinity between the miRNA and target sequence [80]. The miRNAs duplexes with their respective genes represent miR-145-5p:*APLNR,* miR-145-5p:*NOS1*, and miR-145-5p:*RPS6KB1*. The MFE result of <−30 Kcal/mol retained the lowest MFE value of <−30 Kcal/mol and was therefore predicted to yield a stable and perfect hybrid. Virtually all the miR-DEGs that exhibit a good hybrid with the predicted regulatory miRNA were up-regulated. However, it is a known fact that miRNA reduces the expression of the target gene by interacting with it [35,36]. The reason for the up-regulation of target genes of miR-186-5p (*GNB4* and *IL6ST*) and miR-145-5p (*GNAQ*, *SOCS2*, *RPS6KB1*, *GABARAPL1,* and *NOS1*) could be the up-regulation of hsa_circ_0092576 and hsa_circ_0126672, which suppress the expression of miR-186-5p and miR-145-5p, thereby up-regulating their target genes. Hence the circRNA involved in the disease pathology by ceRNA mechanism. To the best of our knowledge, this is the very first study that mechanistically explains the role of hsa_circ_0126672 in the pathophysiology of CHD. However, microarray data shows up-regulation of the hsa_circ_0126672 in the peripheral blood of CHD patients [25].

Modification of the function of endogenous ncRNA by small molecules could be a promising strategy to achieve efficient treatment for ncRNA-related diseases. As an association between miRNAs and CHD has been predicted in this study, the identification of small molecules that potentially modify the expression of these miRNAs may offer a novel therapeutic approach. In this research, it was revealed that miR-145-5p, miR-203a-3p, and miR-7-5p had a higher degree of interaction with the validated small molecules. It was observed that 5-fluorouracil (5-FU) and goserelin are small molecules that interact with miR-7-5p and miR-203a-3p to inhibit their expression. 5-FU has been used in the treatment of cancer, and its regulatory effect on miRNAs including miR-7-5p and miR-203a-3p has been reported [81]. However, cardio-toxicity of 5-fluorouracil was reported [82,83]. Another small molecule, glucocorticoid, interact with miR-145-5p to enhance their expression and miR-203a-3p to inhibit it. The molecule (glucocorticoid) has been used widely in the treatment of rheumatic diseases and is a potent anti-inflammatory drug [84]. Its anti-inflammatory effect is evident, suggesting that it may have therapeutic benefits for atherosclerosis and CHD [85].

Based on the current data, we presented a summary model to elaborate on the involvement of has_circ _0126672 and their target miRNAs in the pathology of CHD (Figure 8).

It was predicted that has_circ_0126672 would successfully sponge their target miRNAs, diminishing their impact on the target genes of the Apelin and JAK/STAT signaling pathways. As a result, the expression of genes was increased, and the pathways were activated. Activation of the JAK/STAT signaling pathway triggers the inflammatory cascade within the vascular environment, which ultimately induces atherosclerosis through the activation of vascular smooth muscle cells (VSMCs) [67,68,86]. Likewise, the activated Apelin signaling pathway facilitated atherosclerosis by promoting oxidative stress in the vascular environment, resulting in the deposition of oxidized lipids and eventually endothelial damage [76,87]. The evolution of atherosclerotic plaque in the coronary artery eventually led to CHD.

## 5. Conclusions

In this study, computational analyses were employed to predict the role of circRNAs and their target miRNAs as potential biomarkers of CHD. hsa_circ_0126672 and its target miRNAs (miR-7-5p, miR-548c-3p, miR-21-5p, miR-186-5p, miR-145-5p, miR-203a-3p, and miR-495-3p) were identified as crucial molecules in the pathogenesis of CHD. It was demonstrated that hsa_circ_0126672 and their target miRNAs are implicated in the pathogenesis of CHD via the regulation of CHD-related cellular functions as well as many relevant signaling pathways. The effect of hsa_circ_0126672 on Apelin and the JAK/STAT signaling pathway was emphasized, and the affected genes were *IL6ST*, *STAT3*, *JAK3*, *APLNR*, *GNAQ*, *GNB4*, *PRKACB*, *NOS1*, *GABARAPL1,* and *RPS6KB1.* Furthermore, we predicted stable and perfect interactions in the duplexes of miR-145-5p:*APLNR*, miR-145-5p:*NOS1*, and miR-145-5p:*RPS6KB1.* Thus, the potential regulatory effect of these miRNAs on their respective target genes was predicted. Small molecules that could interact with the miRNAs to enhance or inhibit their expression were also demonstrated, suggesting a new strategy for the management of the disease through the modification of miRNA expression. Thus, it is suggested that hsa_circ_0126672 along with its target miRNAs could act as potential biomarker panels for the diagnosis and treatment of CHD. However, rigorous wet laboratory experiments are required to further validate these findings.

## Figures and Tables

**Figure 1 genes-14-00550-f001:**
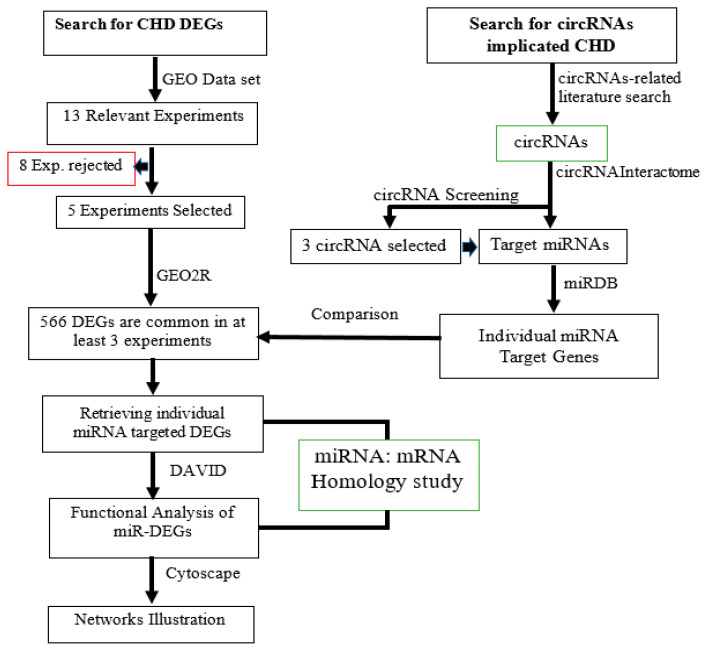
Summary of the workflow used in this study. GEO: gene expression omnibus, CHD: coronary heart disease, DEG: differentially expressed genes, miR-DEGs: differentially expressed miRNA target genes, circRNAs: circular RNAs, miRDB: microRNA database, miRNA: microRNA, DAVID: database for annotation, visualization, and integrated discovery.

**Figure 2 genes-14-00550-f002:**
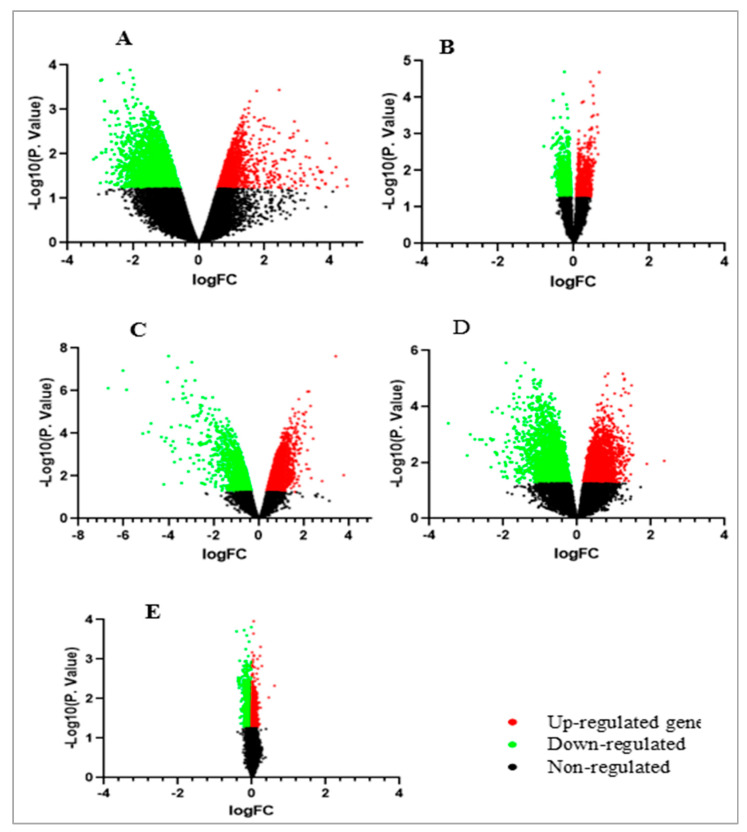
Volcano plot visualizing the differentially expressed genes in CHD microarray datasets: (**A**): GSE71226, (**B**): GSE12288, (**C**): GSE56885, (**D**): GSE42148 (**E**): GSE20681.

**Figure 3 genes-14-00550-f003:**
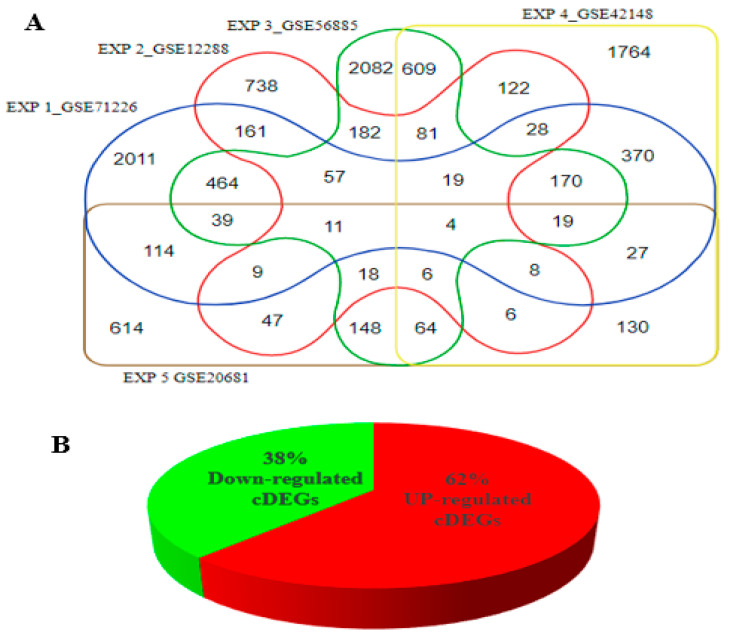
Number of differentially expressed genes (DEGs) implicated in CHD. (**A**): number of genes distributed among the five experiments used in the study. (**B**): percentage composition of up-regulated and down-regulated genes associated with CHD implicated in at least three of the five experiments selected.

**Figure 4 genes-14-00550-f004:**
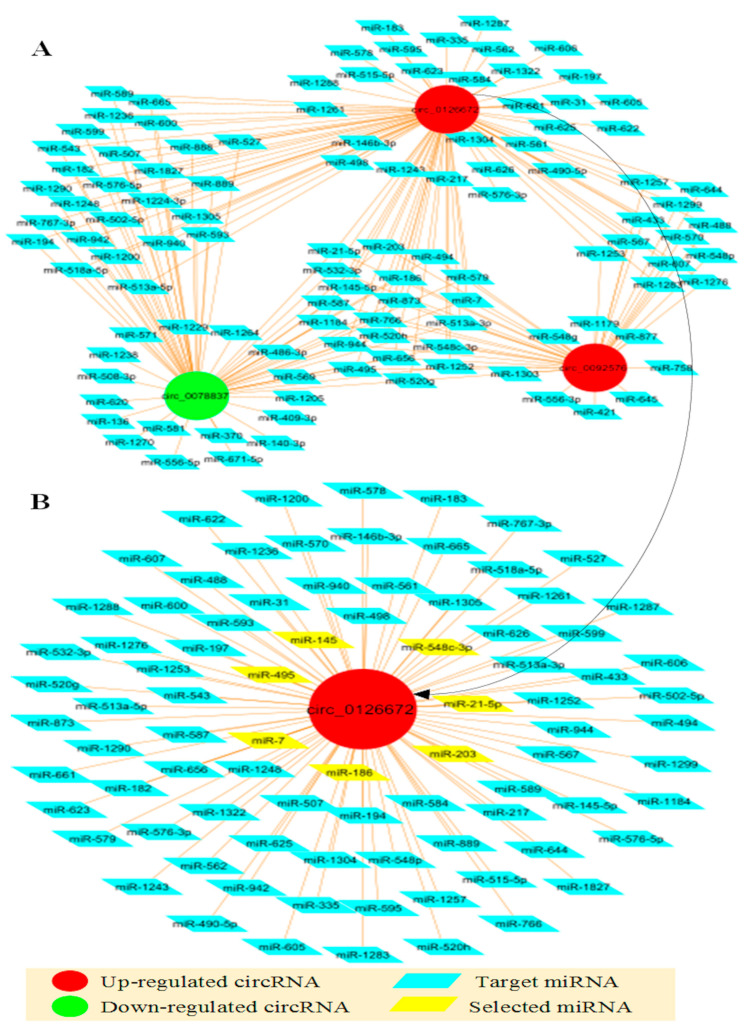
circRNA-miRNA networks. (**A**): Network of CHD dysregulated circRNAs and their target miRNAs. (**B**): Network of the CHD most relevant circRNA and its target miRNA.

**Figure 5 genes-14-00550-f005:**
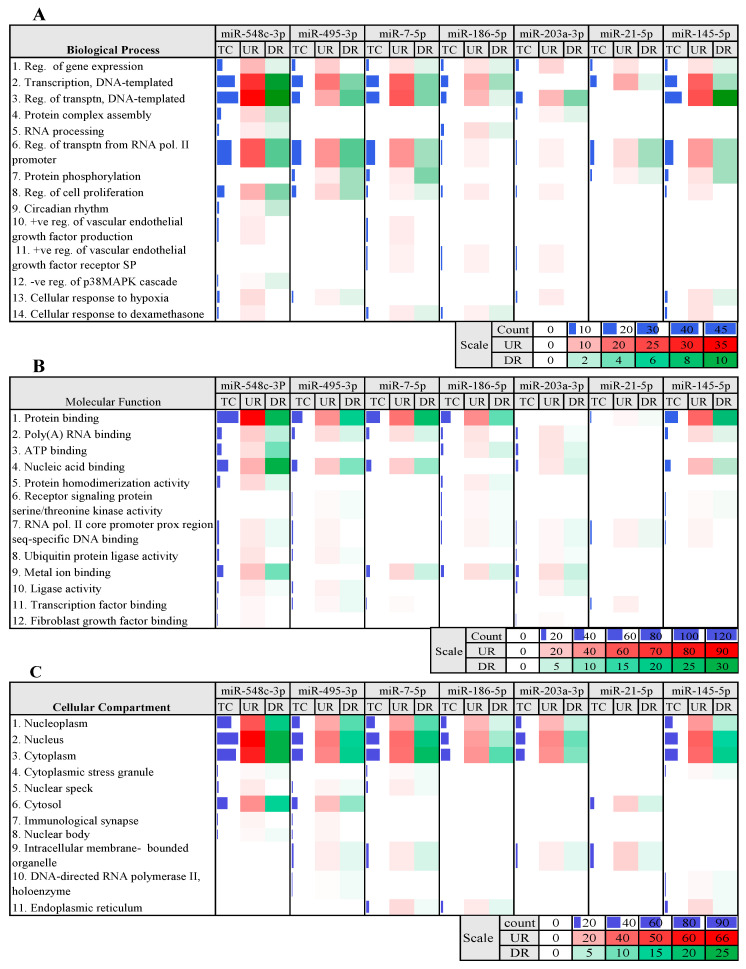
Functional enrichment analyses of miR-DEGs. (**A**): Biological processes associated with miR-DEGs, (**B**): Molecular functions associated with miR-DEGs, (**C**): Cellular compartments of the miR-DEGs. Color intensities show the variation in expression. DR: down-regulated genes, UR: up-regulated genes, Pol II: polymerase II, Seq: sequence, Reg: regulation, +ve: positive, −ve: negative.

**Figure 6 genes-14-00550-f006:**
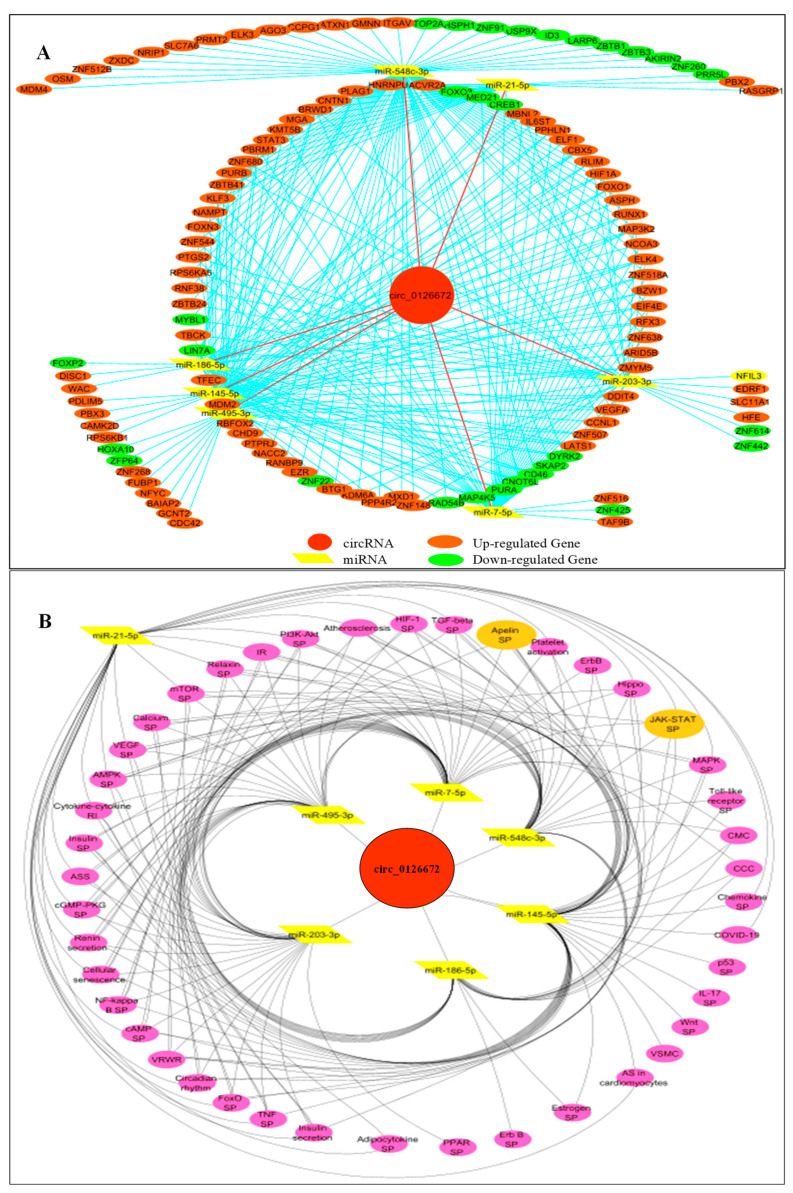
Circular RNA networks. (**A**): hsa_circ_0126672/miRNAs/genes regulatory network in CHD-related biological processes. (**B**): Network interaction between miRNAs and predicted biological pathways affected by the miR-DEGs. SP: Signaling pathway, IR: insulin resistance, RI: receptor interaction, VRWR: Vasopressin-regulated water reabsorption, CMC: Cardiac muscle contraction, ASS: aldosterone synthesis and secretion, CCC: Complement and coagulation cascades, AS: Adrenergic signaling, NAFL: Non-alcoholic fatty liver, VSMC: Vascular smooth muscle contraction.

**Figure 7 genes-14-00550-f007:**
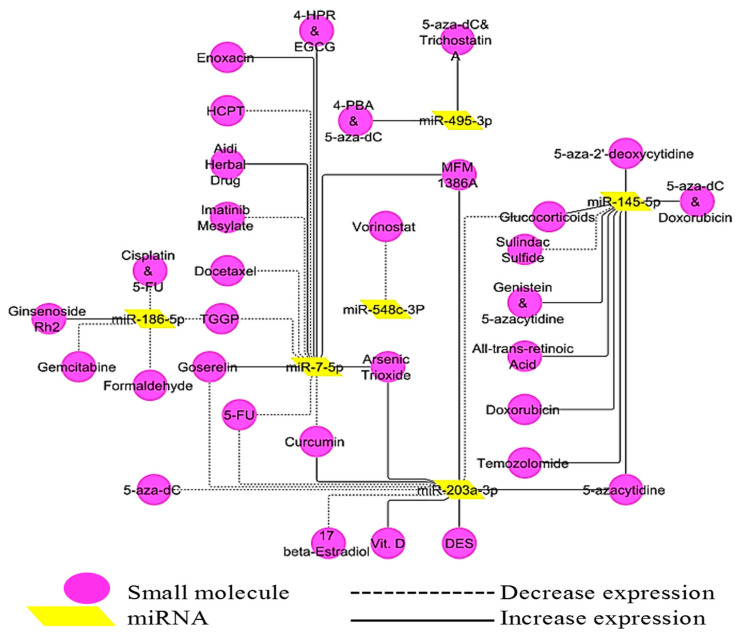
Interaction between miRNAs and small molecules. 4-PBA: 4-phenyl butyric Acid, 5-aza-dC = 5-aza-2′-deoxycytidine, TGGP: 1,2,6-Tri-O-galloyl-β-D-glucopyranose, HCPT: Hydroxycamptothecin, 5-FU: 5-Fluorouracil, 4-HPR: N-(4 Hydroxyphenyl) retinamide, EGCG: Epigallocatechin Gallate, MFM: Marine Fungal Metabolite, Vit. D: vitamin D, DES: Diethylstilbestrol.

**Figure 8 genes-14-00550-f008:**
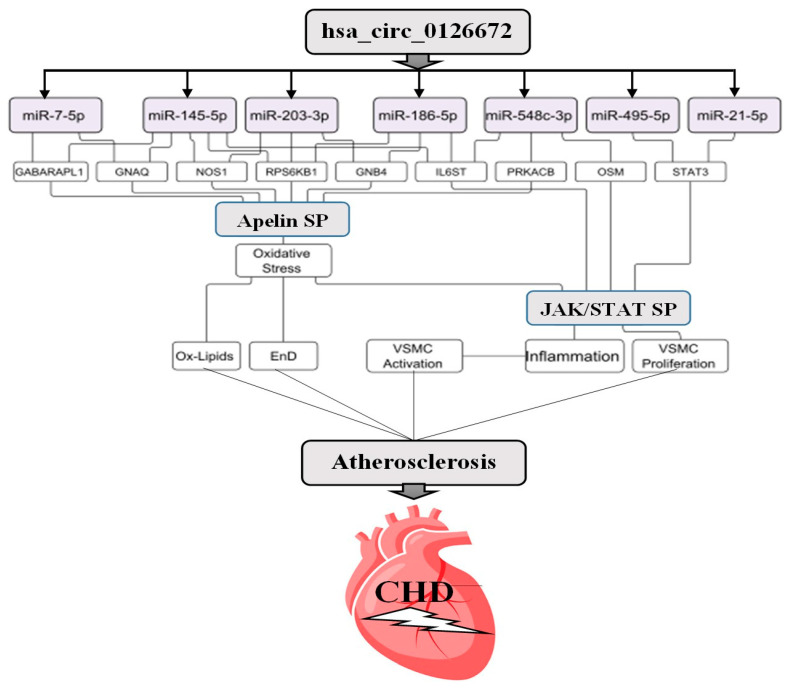
Proposed summary model indicating the role of hsa_circ_0126672 in the pathogenesis of CHD. CHD: coronary heart disease, EnD: endothelial dysfunction, SP: signaling pathway, ox-lipids: oxidized lipid, VSMCs: vascular smooth muscle cells.

**Table 1 genes-14-00550-t001:** Selected experiments and their details.

Accession ID	Gene Expression Platform	Sample Type	Number of Controls	Number of Cases	Total Samples	Location
GSE71226	GPL570 Affymetrix Human Genome U133 Plus 2.0 Array	Peripheral blood	03	03	06	China
GSE12288	GPL96 Affymetrix Human Genome U133A Array	Peripheral blood	112	110	222	Switzerland
GSE56885	GPL15207 Affymetrix Human Gene Expression Array	Peripheral blood mononuclear cells	02	04	06	India
GSE42148	GPL13607 Agilent-028004 SurePrint G3 Human GE 8x60K Microarray	Peripheral blood	11	13	23	India
GSE20681	GPL4133 Agilent-014850 Whole Human Genome Microarray 4x44K G4112F	Peripheral blood	90	90	198	USA

## Data Availability

Not applicable.

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
