# Peer review of "Competing Endogenous RNA Regulatory Networks of hsa_circ_0126672 in Pathophysiology of Coronary Heart Disease"

_genes, 2023, doi:10.3390/genes14030550_

Round 1

Reviewer 1 Report

- Although the study are valuable, the way that the author presents are so confusing and in some parts are vague. 

- The authors need to add that this study is only bioinformatics analysis and for all the data that they presented need to validate by experimental analysis. 

- The authors must reference the recent published papers, all of the paper cited in the introduction part are reported before 2019. 

- They must add all the GSE data set numbers in the methods and explain them what are the samples and why they choose them to analyzed. 

- Why the authors didn't use the logFc as a statistical parameter for  microRNAs with differential expression. 

- The authors must explain clearly about the functional analyses part. Are they only did the analysis for genes and/or microRNAs? for which groups of ncRNAs and/or genes the GO and pathway analysis were done?

- Results 

The results part must be rewrite and restructured. 

The author must add some subtitles and explain clearly in each part what are the outcomes. 

The author must clearly explain why they chose hsa-circ-0126672 for further analysis according to which results. 

Since the authors employed several GSE, are the samples are similar in each data sets? Are the authors need to apply batch effect removal for data? They must explain this. 

Are there any available data about the expression of miRNAs in CHD patients? The authors must confirm their results at least either in available data or experimental analysis. 

What are the p value and logFc for miRNAs target in their analysis? 

- Discussion 

- must restructure according to the restructured results. 

The authors must add probable for all the data that they got since their results are from bioinformatic analysis. 

Are the miRNAs that they introduces as therapeutic targets are associated with the Apelin pathway and its involved genes?

The authors must changed the abstract corresponding to new structured manuscript. 

Author Response

Response to Reviewer

We would like to thank the reviewers and Editor-In-Chief for their valuable comments. We have tried our best to address all comments and concerns.

Comments and Suggestions for Authors:

Reviewer #1:

- Although the study are valuable, the way that the author presents are so confusing and in some parts are vague. 

Author’s response: Thank you for the valuable comments and suggestions. We have updated the manuscript to minimize the confusion.

- The authors need to add that this study is only bioinformatics analysis and for all the data that they presented need to validate by experimental analysis. 

Author’s response: We have added the suggestion in the conclusion that this is a bioinformatics prediction and required rigorous wet laboratory analysis in future to validate these findings.

- The authors must reference the recent published papers, all of the papers cited in the introduction part are reported before 2019. 

Author’s response: We have updated the introduction with recent references.

- They must add all the GSE data set numbers in the methods and explain them what are the samples and why they choose them to analyzed.

Author’s response: We have included suggested details in Table 1. We chose these GSE experiments on the basis of a) nature of the samples - peripheral blood b) samples with healthy controls and patients of coronary heart disease (CHD).

- Why the authors didn't use the logFc as a statistical parameter for microRNAs with differential expression. 

Author’s response: We identified miRNAs by submitting the reported circular RNA (has_circ_0126672) to circular RNA Interactome. This circular RNA was abundantly sponging numerous miRNAs; our selected miRNAs had more than ten binding sides on has_circ_0126672 and characteristically targeted a higher number of overlapping differentially expressed genes in 5 gene expression experiments. 

- The authors must explain clearly about the functional analyses part. Are they only did the analysis for genes and/or microRNAs? for which groups of ncRNAs and/or genes the GO and pathway analysis were done?

Author’s response: We have explained in the methodology part that we identified common differentially expressed genes (cDEGs) among all five selected experiments. And miRDB was used to identify miRNA target genes. Then we selected common genes between the miRNA target genes and cDEGs for each miRNA and submitted them to DAAVID for GO.

- Results 

The results part must be rewrite and restructured. The author must add some subtitles and explain clearly in each part what are the outcomes. 

Author’s response: We have updated the result section with sub-headings.

The author must clearly explain why they chose hsa-circ-0126672 for further analysis according to which results. 

Author’s response: We have explained why and how hsa-circ-0126672 was selected for further analysis in the methodology section.

Since the authors employed several GSE, are the samples are similar in each data sets? Are the authors need to apply batch effect removal for data? They must explain this. 

Author’s response: The detail of samples used in this study is listed in table 1. We used GEO2R for the analysis with the selection of healthy control and CHD samples.

Are there any available data about the expression of miRNAs in CHD patients? The authors must confirm their results at least either in available data or experimental analysis. 

Author’s response: The available data for the selected miRNAs have been discussed in our discussion section.

What are the p value and logFc for miRNAs target in their analysis? 

Author’s response: We predicted these miRNAs by submitting our circular RNA to the online database. This database only provides details of predicted miRNAs that can bind to a particular circular RNA without their p-value and logFc. That is the reason we did not mention logFc and p-value.

- Discussion 

- must restructure according to the restured results. 

Author’s response: To minimize the confusion, we have updated our methodology and results parts and discussion is followed by results. However, we have updated the conclusion that this bioinformatics work needs rigorous laboratory analysis.

The authors must add probable for all the data that they got since their results are from bioinformatic analysis. 

Author’s response: We have mentioned it in the manuscript that this is a bioinformatic analysis-based prediction and rigorous wet laboratory experiments are required to validate the finding.

Are the miRNAs that they introduces as therapeutic targets are associated with the Apelin pathway and its involved genes?

Author’s response: We highlighted in the summary model and proposed that these miRNAs could regulate the JAK/STAT and Apelin pathway crucial for the onset and progression of CHD.

The authors must changed the abstract corresponding to new structured manuscript

Author’s response: We have improved the overall manuscript as suggested.

Reviewer 2 Report

In this report, the authors had used a number of existing computer programs to analyze the available data or information which included differential gene expression data and miRNA database, and identified a particular circRNA to be a high target for CHD. 

Improvements on the manuscript may include:

1. Provide the rationale of selecting those five experiments in Fig. 2. and list of some of the key features of those experiments such as the type of array and number of samples etc.

2. For readers with different backgrounds, it would be helpful to explain why those six miRNAs that are highlighted in Fig 4B are located closer to the cirRNA target. If those miRNAs have better scores than the other miRNAs, it would be helpful to include what is the range of scores among all the miRNAs include in Fig 4B.

3. In Fig 7, are all those interactions equal to each other? If not, the variations of interactions should be presented in the figure.

Author Response

Response to Reviewer

We would like to thank the reviewers and Editor-In-Chief for their valuable comments. We have tried our best to address all comments and concerns.

Comments and Suggestions for Authors:

Reviewer # 2

Comments and Suggestions for Authors

In this report, the authors had used a number of existing computer programs to analyze the available data or information which included differential gene expression data and miRNA database, and identified a particular circRNA to be a high target for CHD. Improvements on the manuscript may include:

  1. Provide the rationale of selecting those five experiments in Fig. 2. and list of some of the key features of those experiments such as the type of array and number of samples etc.

Author’s response: We have included suggested details in Table 1. We chose these GSE experiments on the basis of a) nature of the samples - peripheral blood b) samples with healthy controls and patients of coronary heart disease (CHD).

  1. For readers with different backgrounds, it would be helpful to explain why those six miRNAs that are highlighted in Fig 4B are located closer to the cirRNA target. If those miRNAs have better scores than the other miRNAs, it would be helpful to include what is the range of scores among all the miRNAs include in Fig 4B.

Author’s response: We identified miRNAs by submitting the reported circular RNA (has_circ_0126672) to circular RNA Interactome. This circular RNA was abundantly sponging numerous miRNAs; our selected seven miRNAs had more than ten binding sides on has_circ_0126672 and characteristically targeted a higher number of overlapping differentially expressed genes. Hence, we showed these 7 miRNAs closer to the has_circ_0126672. These details are mentioned in the methodology part.

In Fig 7, are all those interactions equal to each other? If not, the variations of interactions should be presented in the figure.

Author’s response: In the footnote of Fig 7, we have mentioned that a solid line represents increased expression while a dotted line represents a decreased expression.

Round 2

Reviewer 1 Report

All comments are addressed except these two following comment which are important:  

1. Since the authors employed several GSE, are the samples are similar in each data sets? Are the authors need to apply batch effect removal for data? They must explain this.

2.  Are there any available data about the expression of miRNAs in CHD patients? The authors must confirm their results at least either in available data or experimental analysis.